# Sonidegib Inhibits the Adhesion of Acute Myeloid Leukemia to the Bone Marrow in Hypoxia: An Optical Tweezer Study

**DOI:** 10.3390/biomedicines13030578

**Published:** 2025-02-25

**Authors:** Katarzyna Gdesz-Birula, Sławomir Drobczyński, Krystian Sarat, Kamila Duś-Szachniewicz

**Affiliations:** 1Department of Clinical and Experimental Pathology, Institute of General and Experimental Pathology, Wrocław Medical University, 50-368 Wrocław, Poland; 2Department of Optics and Photonics, Faculty of Fundamental Problems of Technology, Wrocław University of Science and Technology, 50-370 Wrocław, Poland; slawomir.drobczynski@pwr.edu.pl; 3Laboratory of Genetics and Epigenetics of Human Diseases, Department of Experimental Therapy, Ludwik Hirszfeld Institute of Immunology and Experimental Therapy, Polish Academy of Sciences, 53-114 Wrocław, Poland; krystian.sarat@hirszfeld.pl

**Keywords:** acute myeloid leukemia, optical tweezers, 3D cell culture, bone marrow microenvironment, sonidegib

## Abstract

**Background:** Acute myeloid leukemia (AML) is a heterogeneous disease highly resistant to chemotherapeutic agents. Leukemia stem cells (LSCs) can enter a dormant state and avoid apoptosis in the protective niche of the bone marrow (BM) microenvironment. Moreover, bone marrow stromal cells protect leukemia cells by promoting pro-survival signaling pathways and drug resistance. Therefore, attenuating interactions between leukemia cells and BM cells may have a positive therapeutic effect. **Objectives:** In this work, we hypothesized that sondages may inhibit the adhesion of leukemia cells to the bone marrow by inhibiting the Hedgehog (Hh) signaling pathway. The Hedgehog pathway is a key therapeutic target in AML due to its role in leukemic cell growth and survival. **Methods:** We investigated the effects of sonidegib on the adhesion of individual OCI-AML3 cells to a bone marrow stromal spheroid derived from the HS-5 cell line. For this purpose, we precisely determined the minimum cell-to-cell adhesion time using optical tweezers under normoxic (21% of O_2_) and hypoxic (1% of O_2_) conditions. **Results:** Our results demonstrated that sonidegib significantly increased the minimum cell-to-cell adhesion time necessary for leukemic cells to establish adhesive bonds with bone marrow stromal cells, thereby indicating a reduction in their adhesive properties. Additionally, we showed that sonidegib is particularly effective at hypoxic oxygen concentrations. **Conclusions:** The results obtained in this study suggest that sonidegib, through its modulation of the Hedgehog signaling pathway, holds promise as a potential therapeutic approach to target leukemic cell adhesion within the bone marrow microenvironment.

## 1. Introduction

Acute myeloid leukemia (AML) is a genetically heterogeneous disease with an unfavorable prognosis under the current standard of care [1]. Many patients do not qualify for intensive chemotherapy due to their advanced age. However, for targeted leukemia therapies, the percentage of complete remission is inhibited by minimal residual disease (MRD), leading to a disease relapse [2]. Minimal residual disease (MRD) is the presence of a population of leukemic cells that have survived the treatment and whose numbers do not cause clinical symptoms in the patient. MRD often includes leukemia stem cells that survive in a protected niche in the bone marrow (BM) environment [3,4]. Leukemia stem cells (LSCs) represent a subpopulation of leukemic blasts responsible for disease relapse after a period of remission. In acute myeloid leukemia (AML), these cells preferentially localize to the endosteal region of the bone marrow, where they engage in interactions with various stromal components, including bone marrow mesenchymal stromal cells (MSCs). Studies indicate that BM MSCs are directly involved in the chemotherapy resistance of hematological malignancies, including AML, protecting them from drug-induced apoptosis [5,6,7]. Weakening the interactions between leukemia cells and the BM may, therefore, be beneficial to the patient’s treatment course by freeing leukemia cells from the protective niche of the bone marrow.

One of the key pathways facilitating the interaction between leukemia cells and bone marrow stromal cells is the Hedgehog (Hh) signaling pathway [8,9]. This pathway is crucial in regulating cell growth and differentiation, initiated when Hh ligands—Sonic, Desert, or Indian Hedgehog—bind to the Patched (PTCH) receptor. This interaction relieves PTCH’s inhibitory effect on the Smoothened (SMO) protein, allowing SMO to activate downstream GLI transcription factors [8,9,10,11,12]. Notably, the expression levels of components of the Hh pathway are significantly elevated in patients with acute myeloid leukemia (AML) compared to healthy individuals [8].

Research indicates that inhibiting the Hh pathway in AML not only exhibits anti-tumor activity and enhances the efficacy of existing therapies [10,13] but also diminishes cell adhesion [8,14]. However, studies directly linking the Hh pathway to reduced cell adhesion in AML are still lacking. Furthermore, paracrine signaling through the Hh pathway by bone marrow stromal cells has been implicated in various hematological malignancies [8,11,14,15]. For instance, Dierks et al. [11] demonstrated that inhibiting SMO protein reduced the number of LSCs in vivo and prolonged cancer remission. In a separate study [15], the same authors observed that the in vitro survival of lymphoma partially depends on Hh ligands secreted by stromal cells. Similarly, Hedge et al. [9] found that Hh signaling induced by stromal cells enhances the survival of B-cell chronic lymphocytic leukemia (B-CLL) cells, and this pro-survival effect is abolished by Hh inhibitors.

Wellbrock’s study [8] further corroborated these findings, revealing a significant inhibitory effect on the colony-forming capacity of primary AML cells and AML cell lines following the blockade of the Hh pathway. Additionally, Zhang et al. [14] identified sonidegib, a potent SMO inhibitor, as a selective inhibitor of migration and adhesion in mantle cell lymphoma (MCL) cells to the bone marrow. This inhibition occurs through mechanisms such as the VLA-4-mediated inactivation of focal adhesion kinase (FAK) signaling and also modulates the cytokine and chemokine profiles of the bone marrow stromal cells.

Given that adhesive interactions have been observed between AML cells and the bone marrow’s stromal cells, we hypothesized that sonidegib could impair the adhesion of AML cells and disrupt their supportive microenvironment, which is critical for leukemia progression and treatment resistance. Consequently, this study aimed to investigate whether sonidegib could reduce the adhesion of OCI-AML3 leukemia cells to a bone marrow stromal spheroid.

Sonidegib, approved by the FDA in 2015 as an oral treatment for basal cell carcinoma (BCC) [16], is also being investigated for its potential in treating hematologic malignancies [17,18,19,20,21,22,23]. In a Phase 1/1b clinical trial, the safety and efficacy of sonidegib combined with 5-Azacytidine were evaluated in patients with newly diagnosed or refractory myeloid malignancies, including AML. The side effects were generally mild, and the median overall survival in the study was 7.6 months [24]. Given its favorable safety profile and promising preclinical efficacy, we chose sonidegib to explore its potential to impair AML cell adhesion within the bone marrow microenvironment.

To mimic the leukemia–stromal cell interaction, we used the HS-5 mesenchymal stromal cell line. However, instead of culturing HS-5 cells as a monolayer in traditional 2D cultures, we utilized them in the form of 3D spheroids. Cells growing in spheroids experience varied access to oxygen, nutrients, and test compounds, depending on their location within the structure [25,26]. This setup mirrors the conditions naturally occurring in the bone marrow, where nutrient, drug, and oxygen supply depend on the distance from the blood vessels [27,28]. Studies have shown differences in drug sensitivity between cells grown in 3D cultures and those grown as monolayers [29]. Therefore, conducting studies with three-dimensional structures is critical. Our team previously developed the 3D culture method with HS-5 bone marrow stromal cells and B-cell lymphomas [30,31]. We observed that the lymphoma and bone marrow cells did not form a uniform sphere, but rather a layered spheroid, with bone marrow cells concentrated at the center and surrounded by successive layers of lymphoma cells [30]. We concluded that the 3D co-culture of HS-5 stromal cells and hematologic cancer cells closely mimics in vivo conditions in the bone marrow [30,31]. We later recreated this mixed spheroid model under controlled conditions, using optical tweezers to combine lymphoma and stromal cells, successfully transferring them to the culture [31]. Building on this experience and the existing literature, we created a 3D model using OCI-AML3 cells and HS-5 cells, as direct contact with HS-5 cells has a protective effect on AML cells [5]. We conducted this study under hypoxic conditions to more accurately simulate the in vivo environment. The partial pressure of oxygen (ppO_2_) in the bone marrow is significantly lower than in other tissues [27], with BM cells distributed along an oxygen gradient that ranged from 5% near blood vessels to as low as 0.1% O_2_ in more distant regions [27]. Our previous research demonstrated a significant effect of oxygen concentration on the adhesive properties of B cells and primary lymphocytes [32]. However, traditional in vitro experiments are often conducted in environments that deviate significantly from physiological conditions, exposing cells to oxidative stress, which can alter their phenotype [33,34,35,36]. Cells cultured under low oxygen conditions (1% O_2_) demonstrated reduced cell-to-cell adhesion compared to those maintained at 21% oxygen [33]. These findings suggest that studies conducted under hypoxic conditions, reflecting oxygen levels in the living organism, may yield results more representative of the in vivo environment.

In this study, adhesion forces were measured using optical tweezers (OTs). The use of optical tweezers allows for precise control over the initiation of the adhesive connections between individual cells on a second-scale timeframe [33,37]. OTs enable the real-time analysis of the effects of compounds on single cells, eliminating the influence of external factors that can affect the results of adhesion tests. In contrast to bulk methods, where cell populations are tested without full control, optical tweezers allow for the selection of cells with similar morphology and size, as well as the exclusion of dead or compromised cells [30,31,32]. Traditional cell adhesion assays used in laboratories typically measure population-level effects over minutes or hours and cannot capture the subtle, real-time changes in cell adhesion as accurately as OTs. Xue Gou’s research team has previously applied optical tweezers to study the adhesion and migration of leukemia cells as well as their interactions with stromal cells [38,39,40]. Particularly relevant to our research is the work in which they investigated disruptions in cell adhesion between the Molm13 leukemia cell line and M2-10B4 cells under the influence of AMD-3100, using optical tweezers, among other techniques [39]. Our team has also successfully utilized this method to study the effects of various compounds on the minimal cell-to-cell adhesion time [30,31,32]. In this experiment, leukemia cells were optically trapped and transferred to stromal spheroids, where they were held in direct contact until a stable junction was formed.

The objective of this study was to investigate the effect of sonidegib on the adhesion of OCI-AML3 leukemia cells to the bone marrow stromal spheroid under hypoxic and normoxic conditions, using optical tweezers to measure real-time changes in cell adhesion (Figure 1). We demonstrated a significant effect of hypoxia on the reduced rate of adhesion between OCI-AML3 leukemia cells and the HS-5 spheroid, as well as the impact of sonidegib in reducing adhesion formation under both hypoxic and normoxic conditions. In untreated cells, the time required for leukemia cells to attach to the HS-5 spheroid was shorter under atmospheric oxygen levels. However, sonidegib significantly prolonged the contact time needed for adhesion between leukemia and the bone marrow stromal cells’ spheroid under both oxygenation conditions. Notably, the anti-adhesive effect of sonidegib was more pronounced at an oxygen concentration of 1%. Based on the findings of this study, sonidegib demonstrates potential as a therapeutic agent for modulating leukemia cell adhesion, particularly under hypoxic conditions, which may contribute to overcoming leukemia’s resistance to therapy and improve treatment outcomes.

## 2. Materials and Methods

### 2.1. Cells and Cell Line Culture

The human bone marrow cell line HS-5 was obtained from the American Type Culture Collection (ATCC, Manassas, VA, USA). The OCI-AML3 cell line was sourced from the German Collection of Microorganisms and Cell Cultures (DSMZ, Braunschweig, Germany). The OCI-AML3 cell line was originally derived from the peripheral blood of a 57-year-old male diagnosed with acute myeloid leukemia (AML FAB M4) in 1987; these cells carry an NPM1 gene mutation (type A) and the DNMT3A R882C mutation. The cells were cultured in the RPMI-1640 medium (Gibco, Paisley, UK) supplemented with 10% fetal bovine serum (Gibco, Paisley, UK) and 1% penicillin–streptomycin (Gibco, Paisley, UK). The cell cultures were maintained at 37 °C in a humidified atmosphere containing 5% CO_2_.

### 2.2. AlamarBlue Assay

Sonidegib (LDE-225) was purchased from Sigma-Aldrich (Steinheim, Germany). LDE-225 working stocks (10 mM) were prepared in culture media and stored at −20 °C. The test was performed to determine the IC_50_ value of the compound for the OCI AML3 cell line under two different oxygen conditions: atmospheric (21%) and hypoxic (1%). Cells were treated with LDE-225 at six concentrations ranging from 0.001 to 100 μM. After adding the test compound, cells were incubated at 21% or 1% oxygen for 48 h in multigas incubators. Subsequently, alamarBlue (Thermo Fisher Scientific, Karlsruhe, Germany) was added at 10% of the total volume, and the plates were reincubated for 24 h at 37 °C under various oxygen concentrations. Absorbance was measured at 570 nm and 630 nm using a spectrophotometer (BioTek Instruments, Winooski, VT, USA). Cell proliferation was assessed by calculating the percentage reduction in alamarBlue using Bio-Rad’s alamarBlue colorimetric calculator [Bio-Rad colorimetric and fluorometric calculator. Available at: https://www.bio-rad-antibodies.com/colorimetric-calculator-fluorometric-alamarblue.html. accessed on 20 March 2024]. IC_50_ values were calculated from a sigmoidal dose–response curve (variable slope) using the GraphPad Prism 9 software (GraphPad Software, San Diego, CA, USA). Further details are available in Appendix A. The concentration of sonidegib of 10 μM was chosen for cell treatment. It was in accordance with previous studies, such as Zhang et al., which demonstrated its efficacy in inhibiting hematological cancer cell adhesion and migration [14].

### 2.3. Preparation of Stromal Cell Spheroids

Spheroids were prepared in agarose hydrogels according to the manufacturer’s instructions, using mold No.12–256 from the commercially available PetriDish^®^ 3D system (Microtissues Inc., Providence, RI, USA). A total of 3.2 × 10^4^ HS-5 cells in 190 µL of medium were seeded per gel (approximately 125 cells per well). Spheroids were grown under normoxic (21% oxygen) and hypoxic (1% oxygen) conditions at 37 °C in 5% CO_2_ for 24 h, after which sonidegib was added to the culture medium. Stromal spheroids grown in the medium without sonidegib were used as controls. The morphology and size of the spheroids were monitored every 24 h using an Olympus IX73 inverted microscope (Olympus, Hamburg, Germany) and the Olympus Cell^A software (version 4.3). After 48 h of incubation with LDE-225, the spheroids were used to measure the adhesion time using optical tweezers.

### 2.4. Preparation of OCI-AML3 Cells

A total of 5.0 × 10^5^ cells per well were seeded onto 6-well plates and grown at 37 °C in a humidified atmosphere of 5% CO_2_, under either 21% or 1% oxygen conditions for 24 h, followed by the addition of 10 µM LDE-225 dissolved in culture medium. Control cells were grown in the medium without the tested compound. Cells were monitored every 24 h using an Olympus IX73 inverted microscope (Olympus, Hamburg, Germany) with the Olympus Cell^A software. After 72 h, the cells were subjected to experiments in OTs.

### 2.5. Live/Dead Staining

HS-5 spheroids and OCI-AML3 cells were additionally tested for viability using the live/dead cell viability/cytotoxicity assay (Thermo Fisher Scientific, Germany). For this purpose, the culture medium was removed from the surface of the hydrogel, and the gels were washed with PBS for 5 min. OCI-AML3 cells were centrifuged and resuspended in a PBS solution three times. A total of 8.3 × 10^4^ OCI-AML3 cells were suspended in 20 µL of a staining solution, and 200 µL of the solution was applied to the surface of the gels. Spheroids in gels and OCI-AML3 cells were incubated with the staining solution for 15 min, following the manufacturer’s instructions. The stained leukemia cell suspension was then transferred to a cell counting slide (NanoEntek, Seoul, Republic of Korea) to visualize the cell monolayer. Green and red fluorescence was detected at excitation/emission wavelengths of 485/530 nm and 550/590 nm, respectively, and imaged using an Olympus BX43 fluorescence microscope with the Olympus cellSens software (version 4.3).

### 2.6. Optical Tweezers System

The adhesion of individual leukemia cells to the bone marrow stromal cells was assessed using optical tweezers, as previously described [31,32,33]. The OT system allows for the manipulation of single cells and enables the real-time observation of the entire process. The optical tweezers were designed and developed in the Laboratory of Optical Manipulation at Wrocław University of Technology, using an Olympus IX71 inverted biological microscope (Olympus, Hamburg, Germany). The optical trap was generated using a 1064 nm Nd: YAG laser (maximum output power of 4W), and the position of the trap within the sample was controlled using a Galvano-mirror XY scanning system. Illumination was provided by high-power LEDs with a peak wavelength of 490 nm, and imaging was captured using a CMOS camera (MC1362, Mikrotron GmbH, Unterschleißheim, Germany).

#### Cell Adhesion Assay with Optical Tweezers

After 72 h, the stromal spheroid was transferred from the agarose mold to an uncoated glass-bottomed dish. Approximately 1 × 10^3^ OCI-AML3 cells in 100 µL of culture medium were then added. Using optical tweezers, a single leukemia cell was captured. The optical trap was used to bring the leukemia cell into close proximity to the surface of the stromal spheroid. The minimum cell-to-cell adhesion time was measured from the moment when contact between the AML cell and the HS-5 spheroid was initiated until the formation of an adhesive bond. To confirm the nascent adhesion formation, three attempts were made to detach the leukemia cell from the stromal spheroid. If the cell contact was disrupted, the leukemia cell was repositioned near the stromal spheroid surface, and the contact time was extended.

### 2.7. Cell Adhesion Washing Assay

HS-5 stromal cells were plated onto 96-well plates at a density of 8 × 10^4^ cells per well and grown under 1% or 21% oxygen for 48 h, reaching a confluence of 90%. Simultaneously, OCI-AML3 cells were cultured in suspensions under 1% and 21% oxygen conditions. After 24 h, LDE-225 was added to the test samples at a final concentration of 10 µM, while medium was added to the control samples. After 48 h, OCI-AML3 cells were seeded onto 96-well plates coated with HS-5 cells at a density of 4 × 10^4^ cells per well and incubated under 1% or 21% oxygen for 10, 30, and 60 min. After each time point, the wells were washed three times with warm PBS, and the MTT solution was added at a final concentration of 0.5 mg/mL. Following 3 h of incubation at 37 °C, the absorbance was measured at 490 nm. The absorbance in the control group of HS-5 stromal cells (without OCI-AML3) was considered to be 100%. The percentage of leukemia cells adhering to stromal cells (BS) at time points of 10, 30, and 60 min was calculated using the formula BS (%) = (Experimental OA/Control OA) × 100.

### 2.8. Statistical Analysis

Statistical analyses were conducted using Microsoft Excel 2021 (Microsoft, Redmond, WA, USA). Differences in cell adhesion following drug treatment were assessed using Student’s *t*-test. A *p*-value < 0.05 was considered statistically significant. Results are presented as the mean ± standard deviation (SD).

## 3. Results

### 3.1. Live/Dead Cell Staining Viability Test: Preparing for an Experiment with Optical Tweezers

To test the effect of sonidegib on OCI-AML3 cell and HS-5 spheroid viability, we established the spheroid co-culture as described in Materials and Methods and, 24 h later, treated the cultured cells with the optimized dose of sonidegib. Forty-eight hours later, a live/dead cell viability/cytotoxicity assay was performed. The results are shown in Figure 2. Live cells stained with green-fluorescing calcein–AM constituted the majority in both cell types, while dead cells, labeled with red-fluorescing ethidium homodimers, accounted for a small percentage of cells. No regions with a pronounced accumulation of dead cells were detected (Figure 2A). The live/dead fluorescent staining revealed that, despite treatment with 10 µM sonidegib for 72 h, no significant impact on cell condition was observed. Sonidegib did not affect the survival of OCI-AML3 cells (Figure 2B) or HS-5 spheroids (Figure 2C). These findings indicate that treatment with 10 µM sonidegib, even after prolonged exposure, does not compromise either the viability of OCI-AML3 cells or HS-5 spheroids, thereby ensuring the reliability of measurements performed using optical tweezers.

### 3.2. The Influence of Sonidegib Treatment and Oxygen Concentration on Single-Cell Adhesion in an Optical Tweezer Experiment

Our research team undertook two tasks. The first task was to investigate the effect of oxygen concentration on the adhesion of OCI-AML3 cells to the HS-5 spheroid. We demonstrated a significant impact of hypoxia (1% O_2_) on the increased rate of minimum cell-to-cell adhesion time (*p* = 0.0018), Figure 3A. Our study showed that the contact time for the adhesion of OCI-AML3 cells to the HS-5 spheroid in normoxia was 11.75 ± 3.85 s, whereas in hypoxia, the mean time was 15 ± 5.06 s. In both conditions, cell-to-cell adhesion established within the time range of 10–20 s. However, in normoxia, 10 s was the most frequent contact time for adhesion to occur in 82.5% of cells, while in hypoxia, OCI-AML3 cells adhered to HS-5 spheroids within 10 or 20 s with equal frequency (Figure 3B).

Next, we examined the effect of sonidegib, an adhesion-lowering compound, on the formation of adhesion under hypoxic and normoxic conditions. The culture of HS-5 spheroids and OCI-AML3 cells was established in a manner analogous to the study of oxygen levels’ effects on cell adhesion. Cells were cultured under either 21% or 1% oxygen for 24 h, after which 10 µM LDE-225 was added. Control cells were cultured in medium without the compound. Forty-eight hours after the addition of sonidegib, cells and spheroids were ready for examination using optical tweezers. The results show that the compound significantly prolonged the minimum contact time required to form an adhesive connection between leukemia cells and the bone marrow stromal spheroid in both oxygenation conditions (Figure 3A). Sonidegib inhibited the adhesion of cells growing in normoxia by an average of 6 s (0.66 times slower) and in hypoxia by an average of 11 s (0.58 times slower) compared to non-treated cells (Figure 3B). The average contact time for sonidegib-treated cells was 17.75 ± 10.97 s and 26 ± 11.94 s in normoxia and hypoxia, respectively. Therefore, the compound is particularly effective at an oxygen concentration of 1%.

### 3.3. Cell Adhesion Washing Assay

This population test was conducted to compare the results with those obtained from the single-cell adhesion study performed using optical tweezers. Both treated and untreated OCI-AML3 cells where seeded onto 96-well plate with HS-5 cells and incubated under two different oxygen concentrations (21% O_2_ and 1% O_2_). A significant reduction in the number of OCI-AML3 cells attached to the stromal cells under the influence of sonidegib was observed (Figure 4). In normoxia, the significant inhibition of adhesion was already noticeable at a time point of 10 min, increased at 30 min, and reached a maximum at 60 min of incubation (*p* < 0.001 for each time points). In hypoxia, a reduction in adhesion was also observed at 10 min (*p* ≤ 0.001) and 30 min (*p* < 0.01), with a maximum at 60 min (*p* < 0.001).

The results also revealed differences in cell adhesion due to oxygenation alone. In the untreated group, a greater number of leukemia cells attached to HS-5 cells in atmospheric oxygen. This process was already noticeable by the 10th minute of the experiment, but the differences in adhesion became significant at 30 min and 60 min (*p* < 0.001). In the group treated with sonidegib, the decrease in adhesion occurred under hypoxic conditions. The reduction in the number of OCI-AML3 cells attached to the stroma was evident at 30 min (*p* < 0.05) and reached a maximum at 60 min (*p* < 0.001). These findings are consistent with those obtained using optical tweezers.

## 4. Discussion

AML blasts accumulate in the bone marrow, peripheral blood, and other organs; however, their proliferation and progression primarily occur in the bone marrow [41,42,43,44]. These reciprocal interactions between acute myeloid leukemia cells and bone marrow mesenchymal stromal cells result in alterations to their transcriptional profiles [45,46,47,48]. This, in turn, contributes to the transformation of the normal hematopoietic niche into a microenvironment that supports leukemia progression [49,50,51] and also contributes to the protection of AML cells from both spontaneous apoptosis and apoptosis induced by anticancer therapies [52,53]. In order to reproduce the interactions of mesenchymal stromal cells with AML cells in the bone marrow in vitro, we used cells of the HS-5 line, which is a widely used model of the hematopoietic microenvironment [54]. The genetic analyses proved their similarity to MSCs in their transcriptional profile, and thus, they can be considered a reliable alternative tool for evaluating the contribution of MSCs in tumor development and immunomodulation [55]. In this study, we used stromal spheroids of the HS-5 cell line to study the direct interaction between bone marrow and leukemic cells. Mesenchymal cells in 3D cultures exhibit behaviors that are more representative of their in vivo counterparts, including changes in morphology, gene expression, and signaling pathways [56,57,58]. For instance, cell polarity and tissue architecture are often better maintained in a 3D environment [57,59]. It is worth mentioning that our team had previously developed the method of co-culture of HS-5 stromal cells with non-Hodgkin lymphoma cells to produce hybrid spheroids, which closely mimic the in vivo interactions between lymphoma cells and their microenvironment [30].

Modern therapies in hemato–oncology rightly focus on interfering with the tumor microenvironment due to the proven influence of the microenvironment on the chemoresistance of leukemia cells [56,60]. The disruption of receptor–adhesion molecule interactions between MSC niches and leukemic cells seems to be a good therapeutic target [52,53]. Sadovskaya et al. demonstrated that as many as 137 proteins are secreted in significantly higher amounts by MSCs in patients with acute myeloid leukemia, compared to healthy individuals. Among these, proteins crucial for cell adhesion, such as CD44, COL8A1, LGALS3, and CXCL12 (SDF-1), are markedly increased [51].

Studies indicate that one of the pathways implicated in cell adhesion in hematological cancers is the Hedgehog (Hh) pathway [8,14]. The Hh signaling pathway is crucial in embryonic development, the maintenance of stem cells, and the processes of tissue regeneration. Conversely, the inappropriate activation of the Hh pathway has been linked to cancer development [13,61,62]. Wellbrock et al. [8], in their work on the Hedgehog (Hh) pathway, observed the expression of the Smoothened (SMO) and Patched-1 (PTCH1) receptors, as well as their downstream mediators, GLI1, GLI2, and GLI3, but did not detect any expression of Hedgehog ligands in AML cells. Furthermore, they reported significantly elevated levels of Desert Hedgehog (DHH) in the plasma of AML patients, suggesting a paracrine delivery of Hedgehog ligands by bone marrow stromal cells. These findings highlight the critical role of the bone marrow microenvironment in AML progression, presenting therapeutic opportunities by targeting these interactions. In the same study, silencing GLI2 and GLI1 expression in AML cells using the GLI inhibitor GANT61 led to a marked increase in apoptosis, reduced cell proliferation, and a diminished colony-forming capacity of AML cells [8]. Sonidegib, the drug used in our study, also belongs to the class of Hh pathway inhibitors; however, it targets SMO rather than GLI. Zhang et al. found that blocking Hh signaling in mantle cell lymphoma (MCL) led to the reduced production of stromal cell-secreted factors such as SDF-1, IL-6, and VCAM-1 (the ligand for VLA-4) [14]. AMLs protect themselves against chemotherapy-related apoptosis by producing adhesion molecules such as VLA-4 which, together with VCAM-1, not only promote proliferation in AML and MSCs via the nuclear factor kappa B (NF-κB) pathway but additionally ensure integration into the vascular niche [63,64,65]. VLA-4 also interacts with fibronectin in the bone marrow, an interaction that is considered a major cause of minimal residual disease [64]. In Zhang’s study, the treatment of bone marrow stromal cells with sonidegib resulted in the complete inhibition of MCL adhesion and migration. Similarly, in our study, we observed that sonidegib significantly prolonged the time required for OCI-AML3 cells to establish adhesive interactions with HS-5 spheroids. We examined the adhesion of AML cells under the influence of sonidegib at a concentration of 10 μM, while the research group led by Zhang investigated the adhesion of mantle cell lymphoma (MCL) using the same compound at concentrations of 10 μM and 30 μM.

In 2013, Gou et al. carried out the first work examining the adhesion forces of single leukemia cells to MSCs using optical tweezers, demonstrating the effectiveness of OTs in measuring the adhesion forces of leukemia cells [38,39]. Previous research performed by our team with the use of OTs concerned the study of lymphoma cell adhesion to MSC cells and was based on an evaluation of the minimum cell-to-cell adhesion time [30,31,32]. This approach offered precise insights into the early stages of interactions between bone marrow and lymphoma cells, and thus, we decided to implement it to study the effect of sonidegib on AML cells’ adhesion.

This study investigated the effects of sonidegib and oxygen concentration on OCI-AML3 cell adhesion to the HS-5 spheroid. The results showed that hypoxia alone significantly inhibits adhesion (*p* = 0.0018), with a mean contact time of 15 ± 5.06 s under 1% oxygen, compared to 11.75 ± 3.85 s in normoxia. For further analysis, cells were incubated with 10 µM sonidegib under both normoxic and hypoxic conditions for 48 h before adhesion measurements. Sonidegib significantly reduced cell adhesion in both conditions, showing a greater effect under hypoxia, which is particularly relevant to in vivo environments with limited oxygen. This finding aligns with our previous study, which indicated significantly reduced adhesive properties in lymphoma cell lines as well as primary cells under hypoxic conditions [32]. Similar conclusions were drawn by the Muz research group in their investigation of Waldenström’s macroglobulinemia, a rare and generally indolent form of non-Hodgkin lymphoma [66]. They indicated that hypoxia diminishes cell-to-cell adhesion, facilitating the escape of lymphoma cells from the bone marrow. Likewise, Azab et al. demonstrated that hypoxia reduces the adhesion of multiple myeloma (MM) cells to the bone marrow, promoting their egress from this compartment [67]. In the future, our findings could be extended to other experimental models, including patient-derived ex vivo cultures, to further explore the effects of sonidegib on leukemic cell adhesion within the bone marrow microenvironment. Additionally, ex vivo experiments utilizing bone marrow samples from AML patients could validate the inhibitory effect of sonidegib on leukemic cell adhesion to mesenchymal stromal cells, providing translational relevance to clinical applications. These approaches would help bridge the gap between in vitro findings and potential therapeutic applications for AML patients.

Although photothermal damage is one of the extensively studied limitations of optical tweezer manipulation on living organisms, we previously examined in detail the impact of our experimental settings on individual lymphoma cells using Trypan Blue [68]. We established that the use of the 1064 nm laser beam with its controlled excitation intensity allows for the safe manipulation of single cells for over 500 s when using a laser output power of 100 mW (with a trap stiffness of approximately 50 pN/µm). In this study, the longest direct exposure of a cell to the laser was 20 s. While we cannot exclude the possibility of changes at the molecular level, our previous work demonstrated the high viability of hybrid spheroids 24 h after manipulations with the optical tweezers [31]. However, cell morphology, as one of the indicators of cell viability, should be constantly monitored during the entire procedure. Cells exhibiting visible protrusions (blebs), irregular shapes, or physical alterations in the plasma membrane (e.g., disrupted cell membrane) should be excluded from the experiment.

Finally, we compared the results achieved with single-cell manipulation using optical tweezers and the 96-well format cell adhesion assay, e.g., the washing assay. Both methods confirmed that sonidegib reduces cell adhesion under hypoxic conditions; however, the minimum cell-to-cell adhesion time recorded with optical tweezers was 10 s, whereas the minimum time required for adhesion to occur with the bulk assay was 10 min. The discrepancies observed between the results obtained based on single-cell and population methods likely arise from their differing sensitivities and temporal resolutions. Bulk washing assays assess overall adhesion strength and cell retention after prolonged contact times (usually ranging from 30 min to a few hours) to ensure sufficient binding of cells to the substrate. In contrast, optical tweezers enable the precise, real-time analysis of the early stages of cell adhesion at the single-cell level, providing critical insights into the minimal time required to establish adhesion, which can range from 5 s to a few minutes. It is clear that optical tweezers complement conventional assays by providing a more detailed understanding of the initial dynamics of cell adhesion.

## 5. Conclusions

Our research has confirmed the inhibitory effect of sonidegib on AML cell adhesion to the bone marrow using the precise tool of optical tweezers. The inhibition of leukemia cell adhesion was even more pronounced in hypoxic conditions, indicating that a reduced oxygen concentration enhances the anti-adhesive properties of sonidegib. This highlights the need for further research on hematologic malignancies within the bone marrow niche under hypoxic conditions.

## Figures and Tables

**Figure 1 biomedicines-13-00578-f001:**
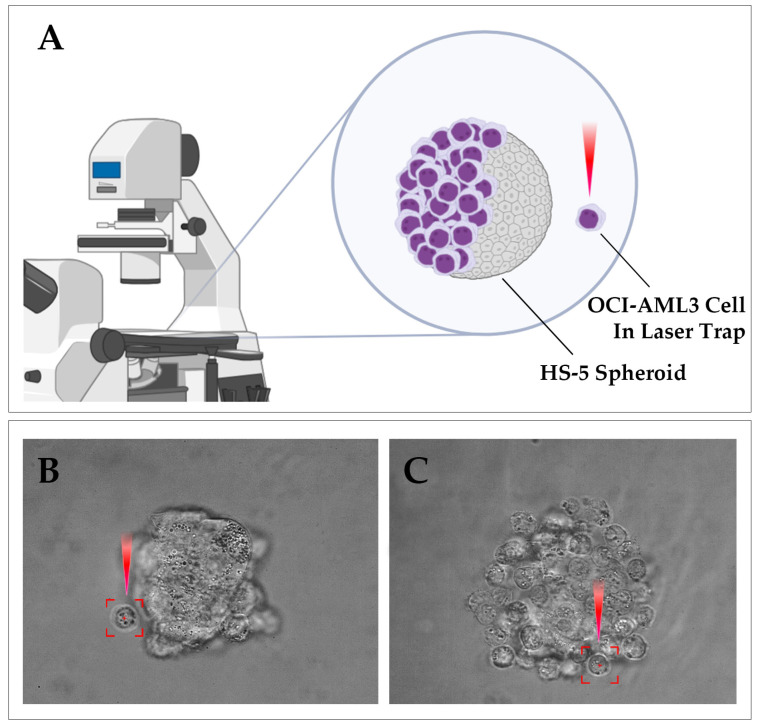
Measurement of adhesion of OCI-AML3 cells to the bone marrow stromal spheroid using optical tweezers (OTs). (**A**) Schematic representation of the experimental setup, using optical tweezers to manipulate the adhesion of individual OCI-AML3 cells to the bone marrow stromal spheroids. OTs allowed for the precise positioning and monitoring of the cell-to-spheroid adhesion process under controlled conditions. (**B**) Representative images of the bone marrow stromal spheroid before and (**C**) after the attachment of the OCI-AML3 cells using OTs. The images, captured during the experiment, show the initial cell-to-spheroid contact, demonstrating successful adhesion.

**Figure 2 biomedicines-13-00578-f002:**
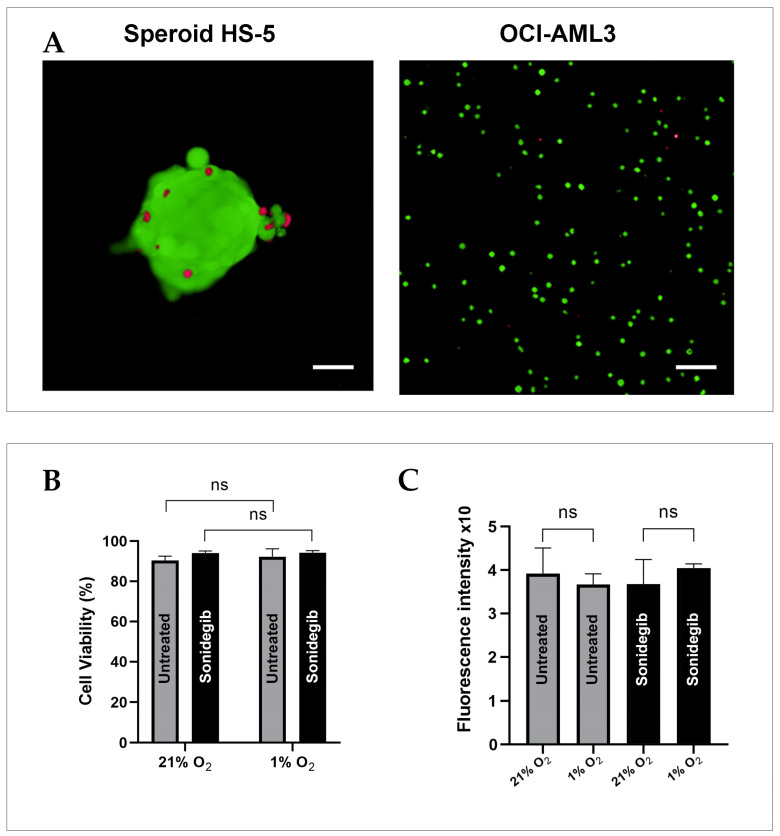
Viability staining of HS-5 spheroids and OCI-AML3 cells following the treatment with 10 µM sonidegib. (**A**) Merged fluorescence microscopy images of HS-5 spheroid (left) and OCI-AML3 cells (right) treated with 10 µM sonidegib for 48 h under hypoxic conditions (1% O_2_). Scale bar = 100 µm. (**B**) Bar graph depicting cell viability (%) based on fluorescence microscopy images from the red and green channels. Data are presented as the mean ± standard deviation (SD) from four technical replicates per condition. Statistical significance was determined using Student’s *t*-test, with error bars representing SD; not statistically significant (ns, *p* > 0.05). (**C**) Fluorescence intensity plots of HS-5 spheroids under two oxygenation conditions. Raw integrated density values from selected regions of interest (ROIs) were normalized to calculate mean fluorescence intensity (MFI). Error bars indicate SD for the analyzed spheroids (n = 4); not statistically significant (ns, *p* > 0.05).

**Figure 3 biomedicines-13-00578-f003:**
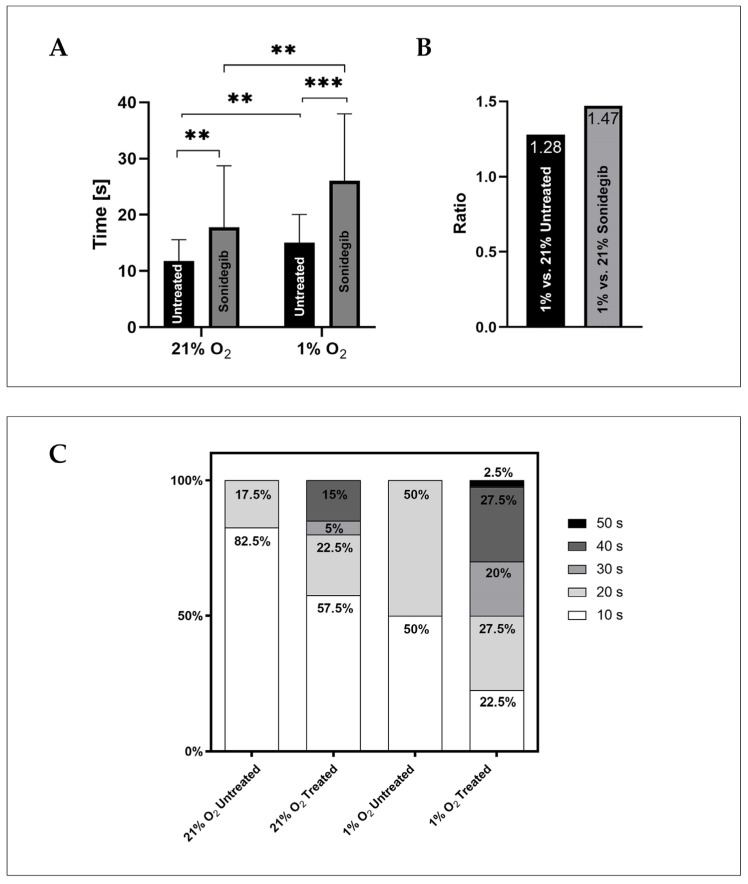
Effect of sonidegib and oxygen concentration on OCI-AML3 cell adhesion to the HS-5 bone marrow stromal spheroid, assessed using optical tweezers (OTs). (**A**) The graph illustrates the average minimum cell-to-cell adhesion time required to establish adhesive connections between OCI-AML3 cells and the HS-5 spheroid. The sonidegib treatment significantly increased the adhesion time by approximately 6 s under normoxic conditions (21% oxygen) and by 11 s under hypoxia (1% oxygen), compared to the control. Hypoxia alone notably extended the minimum cell-to-cell adhesion time, with untreated cells exhibiting a 3.25 s increase in adhesion time under hypoxia versus normoxia and treated cells showing an average extension of 8.25 s. Error bars represent the standard error of the mean (SEM) from two independent experiments (n = 40). ** *p* < 0.01; *** *p* < 0.001. (**B**) The bar graph presents the impact of oxygen concentration on the inhibition of adhesion, expressed as a ratio of the mean adhesion times across different experimental conditions. Data are from two independent experiments (n = 40). (**C**) The bar graph shows the percentage distribution of minimum cell-to-cell adhesion times measured using OTs. Untreated cells typically formed adhesions within 10 to 20 s. Under normoxia, the majority of cells (82.5%) adhered within 10 s, while in hypoxia the distribution was more even, with 50% of cells adhering in 10 s and 50% in 20 s. The sonidegib treatment further prolonged minimum cell-to-cell adhesion times, with some cells requiring up to 50 s to establish connections with stromal spheroid. Data are based on two independent experiments (n = 40).

**Figure 4 biomedicines-13-00578-f004:**
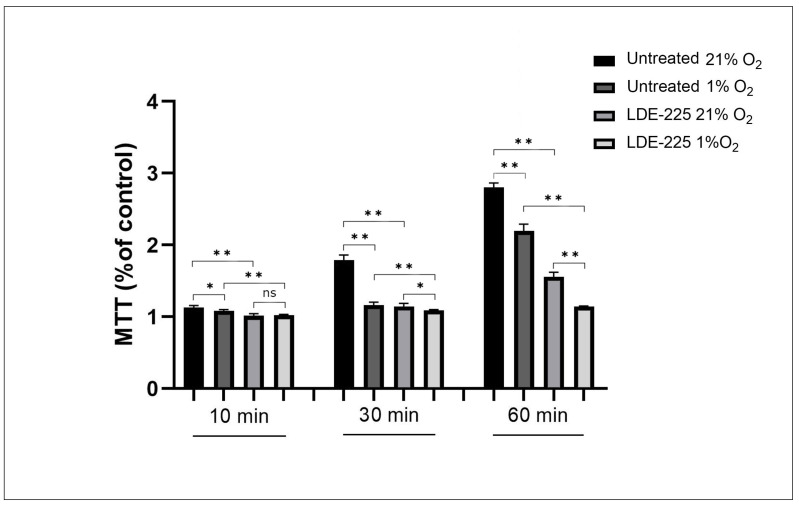
Adhesion of OCI-AML3 cells to HS-5 mesenchymal stromal cells under normoxic and hypoxic conditions. Error bars represent the mean ± standard deviation (SD). Significant differences in leukemia cell adhesion between normoxic and hypoxic conditions are indicated by *p* < 0.05 (*) and *p* < 0.001 (**), while (ns, *p* > 0.05) means not statistically significant, based on statistical analysis.

## Data Availability

The raw data supporting the conclusions of this article will be made available by the authors upon request.

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
