# Peer review of "Sonidegib Inhibits the Adhesion of Acute Myeloid Leukemia to the Bone Marrow in Hypoxia: An Optical Tweezer Study"

_biomedicines, 2025, doi:10.3390/biomedicines13030578_

Round 1

Reviewer 1 Report

Comments and Suggestions for Authors

Dear Editor-in-Chief

Thank you for inviting me to review the revised manuscript titled “Sonidegib affects the adhesion of acute myeloid leukemia to the bone marrow in hypoxia: An optical tweezers study.” In this manuscript, the authors explore the effect of sonidegib on the adhesion of OCI-AML3 leukemia cells to bone marrow stromal spheroids under normoxic and hypoxic conditions, demonstrating its therapeutic capacity in disrupting leukemia-stromal cell interactions. While the study is intriguing, further revisions are required for clarity and scientific rigor.

1. In the Abstract, I suggest changing “one of the drugs which may affect the adhesion of transformed cells by inhibiting the hedgehog (Hh) signaling pathway is sonidegib.” to clarify the meaning.

2. In the Abstract, the sentence "Sonidegib significantly prolonged the contact time required to form an adhesive bond..." could be revised for fluency.

3. In the Introduction, the association between the Hedgehog signaling pathway and AML and the clinical significances could be mentioned earlier.

4. In the Methods, you stated “LDE-225 working stocks (10 mM)”. I want to know why you selected that concentration. Mention the reasons in the manuscript.

5. In the Discussion, I recommend adding a brief discussion of how the findings are able to be validated in vivo or in human samples, which would add weight to the manuscript.

6. A thorough overview of the manuscript is necessary to identify any misunderstandings and errors.

Comments on the Quality of English Language

A thorough overview of the manuscript is necessary to identify any misunderstandings and errors.

Author Response

Dear Reviewer,

We would like to express our sincere gratitude for your invaluable suggestions and constructive feedback. Your comments have significantly improved the quality of our manuscript, and we have carefully addressed all of your points. Below is a summary of the changes made:

  1. In the Abstract, I suggest changing “one of the drugs which may affect the adhesion of transformed cells by inhibiting the hedgehog (Hh) signaling pathway is sonidegib.” to clarify the meaning.

We have revised all poorly phrased sentences in the manuscript and subjected the entire text to a thorough language edit, which was carried out by MDPI, to enhance clarity and readability (Page 1, Line 24).

  1. In the Abstract, the sentence "Sonidegib significantly prolonged the contact time required to form an adhesive bond..." could be revised for fluency.

The sentence in the Abstract has been revised for improved fluency in accordance with your suggestion (Page 1, Line 31).

  1. In the Introduction, the association between the Hedgehog signaling pathway and AML and the clinical significances could be mentioned earlier.

Regarding the section describing the Hedgehog pathway in AML, we placed it at the beginning of the manuscript, in the second and third of the eight introductory paragraphs. However, upon reviewing your insightful comment, we recognized the absence of a discussion on the clinical significance of the Hedgehog pathway in AML within the abstract. This omission has now been addressed and included in the revised version (Page 1, Line 25).

  1. In the Methods, you stated “LDE-225 working stocks (10 mM)”. I want to know why you selected that concentration. Mention the reasons in the manuscript.

Regarding the selection of the 10 μM concentration for the working stock, we would like to thank the reviewer for their valuable comment. In response, we have provided a detailed explanation in the manuscript to clarify our choice (Page 5, Line 203). This concentration was selected based on a previous study by Zhang, which demonstrated significant effects of sonidegib at this concentration in disrupting adhesion and migration of hematological cancer cells. This served as a relevant starting point for evaluating its effects on OCI-AML3 cells in our experimental conditions. Furthermore, we have added the appropriate citation to acknowledge the source of this information.

  1. In the Discussion, I recommend adding a brief discussion of how the findings are able to be validated in vivo or in human samples, which would add weight to the manuscript.

We also appreciate your suggestion to include a discussion on how our findings could be validated in vivo or in human samples. As a result, we have incorporated a section outlining potential experimental models and approaches for validation (Page 13, Line 463). We believe this addition has strengthened the manuscript and further emphasized the clinical relevance of our findings.

  1. A thorough overview of the manuscript is necessary to identify any misunderstandings and errors.

We have commissioned an academic text revision to the MDPI English Language Editing team to ensure accuracy and clarity throughout the manuscript.

Once again, thank you for your thoughtful and helpful suggestions. We believe that the changes made have significantly improved the manuscript.

Best Regards,
Katarzyna Gdesz-Birula

Reviewer 2 Report

Comments and Suggestions for Authors

The goal of this study was to investigate the impact of the PTCH inhibitor effect on the Smoothened (SMO) protein inhibitor sonidegib on the adhesion model of acute myeloid leukemia cells (cell line OCI-AML3) to bone marrow stromal spheroids (HS-5 cell line) under hypoxic and normoxic conditions. Authors used optical tweezers to monitor the changes in cell adhesion in real time.

Sonidegib significantly prolonged the contact time needed for adhesion between leukemia and the bone marrow stromal cells spheroids under both oxygenation conditions. Notably, the antiadhesive effect of sonidegib was more pronounced at an oxygen concentration of 1%

The authors chose a difficult, but almost adequate model to conduct this study. However, it is necessary to take into account that the work was done on cell lines. It is clear that the line HS-5 is very different from the complex hematopoietic microenvironment.

Authors need to coordinate all these phrases and tidy up the text of the article.

Page 3, Line 123. Lower oxygen levels (1% O2) decreased cell adhesion compared to the cells cultured in 21% oxygen. Therefore, it is essential to study the interactions between leukemia cells and BM cells under hypoxic conditions.

Page 8, Line 296. We demonstrated a significant impact of hypoxia on the increased rate of adhesion formation between OCI-AML3 leukemia cells and HS-5 spheroids (p = 0.0018). Apparently, the authors mean adhesion time.

Page 9, Line 323. Sonidegib treatment significantly increased adhesion time by approximately 6 seconds under normoxic conditions (21% oxygen) and by 11 seconds under hypoxia (1% oxygen), compared to the control.

Page 11, Line 435. Results showed that hypoxia alone significantly increased adhesion (p = 0.0018), with a mean contact time of 15 ± 5.06 seconds under 1% oxygen, compared to 11.75 ± 3.85 seconds in normoxia.

Page 12, Line 441. This inhibitory effect on adhesion was most pronounced in hypoxic conditions, where sonidegib extended contact time by 11 seconds on average, compared to 6 seconds in normoxia (p < 0.0001 vs. p = 0.002), in dicating that hypoxia enhances the anti-adhesive properties of sonidegib.

Page 7, Figure 2A Spelling error in the word spheroid.

Comments on the Quality of English Language

The language is almost always understandable, with the exception of the phrases cited in the review.

Author Response

Dear Reviewer,

I would like to express my sincere gratitude for your valuable suggestions and feedback on my manuscript titled „Sonidegib Inhibits Adhesion of Acute Myeloid Leukemia to the Bone Marrow in Hypoxia— An Optical Tweezers Study”. Your insightful comments have greatly contributed to improving the quality of the work, and I truly appreciate the time and effort you dedicated to reviewing it.

I would like to assure you that we have carefully considered all of your remarks and have made the necessary revisions to address the points you raised. All inconsistencies and issues identified in the text have been thoroughly corrected.

  1. Page 3, Line 123. Lower oxygen levels (1% O2) decreased cell adhesion compared to the cells cultured in 21% oxygen. Therefore, it is essential to study the interactions between leukemia cells and BM cells under hypoxic conditions.

The inaccuracies in the cited text fragment have been corrected to ensure clarity and maintain the coherence of the content. We appreciate your careful attention to this detail and believe these adjustments have improved the quality of the manuscript (Page 3, Line 127).

  1. Page 8, Line 296. We demonstrated a significant impact of hypoxia on the increased rate of adhesion formation between OCI-AML3 leukemia cells and HS-5 spheroids (p = 0.0018). Apparently, the authors mean adhesion time.

The cited fragment has been revised to enhance its clarity and ensure that the intended meaning is more easily understood by the reader. We believe this revision improves the overall readability of the text while maintaining its original kontent (Page 8, Line 313).

  1. Page 9, Line 323. Sonidegib treatment significantly increased adhesion time by approximately 6 seconds under normoxic conditions (21% oxygen) and by 11 seconds under hypoxia (1% oxygen), compared to the control.

The cited fragment has been revised to enhance its clarity and ensure that the intended meaning is more easily understood by the reader. We believe this revision improves the overall readability of the text while maintaining its original content (Page 8, Line 313).

  1. Page 11, Line 435. Results showed that hypoxia alone significantly increased adhesion (p = 0.0018), with a mean contact time of 15 ± 5.06 seconds under 1% oxygen, compared to 11.75 ± 3.85 seconds in normoxia.

The indicated fragment has been carefully revised to enhance its clarity and ensure that it is more comprehensible for the reader. We have refined the wording and structure to improve the overall readability and alignment with the context of the text. Thank you for bringing this to our attention (Page 9, Line 331).

  1. Page 12, Line 441. This inhibitory effect on adhesion was most pronounced in hypoxic conditions, where sonidegib extended contact time by 11 seconds on average, compared to 6 seconds in normoxia (p < 0.0001 vs. p = 0.002), in dicating that hypoxia enhances the anti-adhesive properties of sonidegib.

The cited fragment has been revised to improve its clarity and readability. These adjustments were made to ensure that the content is more easily understood and effectively conveys the intended meaning (Page 9, Line 328).

  1. Page 7, Figure 2A Spelling error in the word spheroid.

We have corrected the spelling error in the word ‘spheroid’ at the beginning of the manuscript (Page 8, Line 301). Once again, thank you for your constructive feedback, which has been instrumental in enhancing the clarity and rigor of our research.

Yours sincerely,

Katarzyna Gdez-Birula

Reviewer 3 Report

Comments and Suggestions for Authors

The manuscript investigates the effects of hypoxia and Sonic hedgehog (Shh) signaling pathway inhibitor Sonidegib on the adhesion between acute myeloid leukemia OCI-AML3 cell line and human bone marrow cell line HS-5. The authors have used monolayer and HS-5 stromal cells spheroids to investigate the binding of OCI-AML3 cells by conventional cell adhesion assay and optical tweezers, respectively.  They have found that hypoxic (1% O2) condition and Sonidegib treatment inhibit binding of OCI-AML3 cells to HS-5 stromal cells indicating that targeting Shh signaling pathway is a promising approach for the development of therapeutics for the treatment of leukemia. In general, the manuscript clearly details the findings of the study, and the study appears to be technically sound.

Comments:

1.     Although it is known that Shh pathway inhibitors affect adhesion of cancer cells to bone marrow stromal cells, the present study has applied optical tweezers and used HS-5 cells spheroids to investigate the adhesion between OCI-AML3 and HS-5 cells indicating novel approaches.

2.     Materials and Methods – Provide details about statistical analyses methods.

3.     Optical Tweezers study – i. won’t the laser trap itself damage the cells resulting in artefacts? ii. please define precise positioning of OCI-AML3 cells near stromal cell spheroids so that the experimental parameters are objectively chosen. iii. how did you detach leukemia cell from stromal cell spheroid? Please explain. iv. Fig. 3 legend – is adhesion time same as the contact time? v. Fig. 4 – are you sure the y-axis labeling is correct? According to the data in the figure, a very small percentage (1 – 3%) of OCI-AML3 cells are adhering to HS-5 stromal cells. Is this correct? Please verify. Fig. 4 legend – indicate n = ?.

4.        Discussion section, lines 456 – 458, you write that based on the discrepancies between the optical tweezers study and conventional cell adhesion assay, optical tweezers study provided more detailed information and accurate depiction of mechanisms of adhesion. These conclusions are not supported by data and must be revised or deleted.

Author Response

Dear Reviewer,

We would like to extend our sincerest gratitude for the thoughtful and detailed feedback provided during the review of our manuscript. Your valuable comments and suggestions have greatly contributed to improving the clarity, precision, and overall quality of our work.

Below, we have addressed each of your queries and incorporated the necessary revisions:

1. Statistical Analysis:

We have included the missing statistical analysis section, ensuring that the results are presented with greater accuracy and rigor (Page 7, Line 278).

2. Impact of the Laser on Cells:

In our previous study, we investigated the effect of laser exposure on cells. Our findings demonstrated that laser operation at 1064 nm carries minimal risk of inducing optical damage to primary B-cells. Specifically, both normal B lymphocytes and lymphoma cells were successfully optically trapped for over 10 minutes at a laser power of 100 mW without exhibiting any detectable signs of cellular damage, while fully retaining their trapping stability and motility. Similarly, for OCI-AML3 and HS-5 cells, no cellular damage is expected under these conditions. Therefore, laser manipulation with these settings is considered safe for the cells.

3. Definition of Precise Cell Positioning:Precise cell positioning in our protocol is achieved using optical tweezers in combination with microscope control software. The criteria for precise positioning are based on:

  • Position relative to the spheroid: Each cell is precisely positioned to make direct contact with the spheroid surface, monitored in real-time via microscopic imaging.
  • Use of on-screen markers: Markers and the laser trap position (optical cursor) guide cell movements to exact locations.
  • Focus plane control: Adjustments to the microscope's focal plane allow precise control over cell positioning in both the horizontal and vertical dimensions.

4. Objectivity of Positioning Parameters:
We have ensured the objectivity of positioning parameters by basing them on measurable values:

  • Micrometric control: The optical trap movements are controlled with micrometer precision.
  • Stable contact verification: A stable position is confirmed through a series of detachment attempts using the optical trap.
  • Reproducibility: The process is standardized and repeatable, ensuring consistent results across experiments.

5. Detachment of Leukemic Cells:
We have elaborated on the method for detaching leukemic cells from the stromal spheroid using optical tweezers. The process involves the targeted application of optical force to the attached cell, precise manipulation to assess binding strength, and an objective evaluation of detachment efficiency.

6. Adhesion Time in Figure 3:
We confirm that "adhesion time" and "contact time" are synonymous in our protocol. The minimal cell-to-cell adhesion time is rigorously determined and validated through real-time manipulation and detachment attempts using optical tweezers (Page 10, Line 338).

7. Clarification on Figure 4:
We have verified this and observed similar results in our previous study (Duś-Szachniewicz et al., Int J Mol Sci, 2018). Additionally, we have clarified the cell numbers: 8 × 10⁴ HS-5 cells per well and 4 × 10⁴ OCI-AML3 cells per well (Page 11, Line 374).

8. Discussion Section (Lines 456–458):
We have revised this section (Page 13, Line 463) to emphasize that optical tweezers enable detailed, real-time monitoring of early adhesion events at the single-cell level. While they offer greater precision in observing adhesion processes, they do not necessarily imply higher "accuracy" in describing adhesion mechanisms compared to traditional methods. Specifically:

    • Optical tweezers allow the measurement of minimal adhesion time (10 seconds), which is unattainable with washing assays (10 minutes).
    • The method provides dynamic insights into cell adhesion and eliminates confounding effects from dead or damaged cells.

We hope these revisions address all of your concerns comprehensively. Your constructive input has been instrumental in refining our work, and we deeply appreciate the time and effort you dedicated to reviewing our manuscript. Thank you once again for your invaluable contribution.

Yours sincerely,

Katarzyna Gdez-Birula

Round 2

Reviewer 1 Report

Comments and Suggestions for Authors

The authors addressed my concerns satisfactorily. The manuscript is ready to be published.

Author Response

Dear Reviewer,

We sincerely appreciate your time and effort in reviewing our manuscript. Thank you for your positive assessment and for taking the time to evaluate our work. Your support and feedback are greatly valued.

Best regards,

Katarzyna Gdesz-Birula

Reviewer 3 Report

Comments and Suggestions for Authors

Thank you for responding to my concerns and comments. It would have been helpful to the reviewer if the revised manuscript displayed line numbers and highlighted revised/corrected text (it may not be your fault?). I have a few minor recommendations for correction:

1. Include a sentence indicating that based on your and other published data it is not likely that the viability of cells used in your study may not be affected by optical tweezers.

2. Fig. 4 - the percentage of cells binding in conventional cell adhesion assay is very low (1 - 3%). Can you comment on it? Is it because the assay is not optimized? Although statistically significant differences between treatments are seen, the low binding of cells raises concerns about the validity of the assay.

3. Discussion (page 11, paragraph 1) - Change " In order to reproduce the interactions of mesenchymal stromal cells in the bone marrow in vitro......." to "In order to investigate the interactions of mesenchymal stromal cells with AML cells in vitro....". Reproduce is not correct as you are not studying their interactions in the bone marrow (environment).

4. Discussion (page 13, paragraph 1) - Change "Primary cells under hypoxic (32) to "Primary cells under hypoxic conditions (32).

5. Discussion (page 13, paragraph 2) - Change "single cell manipulation method and bulk assay...." to "single cell manipulation using optical tweezers and 96-well format cell adhesion assay.... "

6. Discussion (page 13, paragraph 2) - Change "...minimal time required to establish adhesive bonds..." to "...minimal time required to establish adhesion.."

7. Discussion (page 13, paragraph 2) - you write "Moreover, the viability of cells is monitored throughout the process...". Please clarify how you monitor the viability of cells? For example, by monitoring the cell shape?

Comments on the Quality of English Language

The quality of English language in the text is ok, however it can be improved to enhance clarity.

Author Response

Dear Reviewer,

Thank you for your careful reading of our manuscript and your valuable comments. We appreciate the time and effort you took to provide your feedback, which has contributed to improving our work. Your insights are greatly appreciated, and we have made the necessary changes based on your suggestions.

  1. Include a sentence indicating that based on your and other published data it is not likely that the viability of cells used in your study may not be affected by optical tweezers.

Although one of the extensively studied limitations of optical tweezers (OT) manipulation on living organisms may be photothermal damage, we previously examined in detail the impact of our experimental settings on living cells, such as using SiμPs microparticles and Trypan Blue (Drobczyński S. and al. Toward Controlled Photothermal Treatment of Single Cell: Optically Induced Heating and Remote Temperature Monitoring In Vitro through Double Wavelength Optical Tweezers. ACS Photonics, 2017, vol. 4, nr 8, s. 1993-2002).

In summary, the 1064 nm laser beam with its controlled excitation intensity was used to intentionally heat the SiμPs while continuously monitoring the surrounding temperature and recording the morphology and condition of the cell in white light mode. This setup allowed for the correlation of temperature and time changes with the condition of the cell. In particular, the accumulation of Trypan Blue (TB) was measured to monitor cell damage resulting from the localized increase in temperature. No TB accumulation was observed in the negative control experiments, indicating that cell death was exclusively attributable to the response of the cell to excessive heating. For more details, please see Figure S4 included in the Supporting Information.

The same conditions were used in our subsequent studies and in the current experiments. In conclusion, the use of the 1064 nm laser beam with its controlled excitation intensity allows for safe manipulations of single cells for over 500 seconds, when using a laser output power of 100 mW (a trap stiffness of approximately 50 pN/µm). In our study, the longest direct exposure of a cell to the laser was 20 seconds. While we cannot exclude the possibility of changes at the molecular level, our previous work demonstrated high viability of hybrid spheroids 24 hours after manipulations were performed with the optical tweezers.

The principles of measuring living cell properties in OT were also discussed in our previous technical papers:  

  1. Drobczyński Sławomir, Duś-Szachniewicz Kamila, Real-time force measurement in double wavelength optical tweezers Journal of the Optical Society of America B-Optical Physics, 2017, vol. 34, nr 1, s. 38-43.
  2. Drobczyński S, …..and Duś-Szachniewicz K. Double wavelength multifunctional optical tweezers. Proceedings of SPIE; nr Vol.10976, ISBN 978-1-5106-2608-9, 2018.
  3. Drobczyński Sławomir, Duś-Szachniewicz Kamila, at al. Spectral analysis by a video camera in a holographic optical tweezers setup Optica Applicata, 2013, vol. 43, nr 4, s. 739-746.

To clarify the issue of photodamage, we propose to include the following sentence in the Discussion section, along with an answer to comment no. 7 (page 14, paragraph 2).

Although photothermal damage is one of the extensively studied limitations of optical tweezers manipulation on living organisms, we previously examined in detail the impact of our experimental settings on individual lymphoma cells using Trypan Blue [68]. We established that the use of the 1064 nm laser beam with its controlled excitation intensity allows for the safe manipulation of single cells for over 500 seconds when using a laser output power of 100 mW (with a trap stiffness of approximately 50 pN/µm). In this study, the longest direct exposure of a cell to the laser was 20 seconds. While we cannot exclude the possibility of changes at the molecular level, our previous work demonstrated high viability of hybrid spheroids 24 hours after manipulations with the optical tweezers were performer [31]. However, cell morphology, as one of the indicators of cell viability, should be constantly monitored during entire procedure. Cells exhibiting visible protrusions (blebs), irregular shapes, or physical alterations in the plasma membrane (e.g., disrupted cell membrane) should be excluded from the experiment.

  1. Fig. 4 - the percentage of cells binding in conventional cell adhesion assay is very low (1 - 3%). Can you comment on it? Is it because the assay is not optimized? Although statistically significant differences between treatments are seen, the low binding of cells raises concerns about the validity of the assay.

The observed low percentage of OCI-AML3 cell adhesion in the conventional assay is consistent with previous findings for B-NHL cells, where adhesion to MSCs did not exceed 0.5%​ [31]. This suggests that the limited binding efficiency is an inherent property of these leukemia and lymphoma cells rather than a result of suboptimal assay conditions.

  1. Discussion (page 11, paragraph 1) - Change " In order to reproduce the interactions of mesenchymal stromal cells in the bone marrow in vitro......." to "In order to investigate the interactions of mesenchymal stromal cells with AML cells in vitro....". Reproduce is not correct as you are not studying their interactions in the bone marrow (environment).

We have replaced "In order to reproduce the interactions of mesenchymal stromal cells in the bone marrow in vitro..." with "In order to investigate the interactions of mesenchymal stromal cells with AML cells in vitro..." as suggested. This revision ensures a more accurate description of the experimental approach and avoids the incorrect implication of reproducing the bone marrow environment.

  1. Discussion (page 13, paragraph 1) - Change "Primary cells under hypoxic (32) to "Primary cells under hypoxic conditions (32).

We have modified "Primary cells under hypoxic (32)" to "Primary cells under hypoxic conditions (32)" to improve clarity and grammatical accuracy.

  1. Discussion (page 13, paragraph 2) - Change "single cell manipulation method and bulk assay...." to "single cell manipulation using optical tweezers and 96-well format cell adhesion assay.... "

We have revised "single cell manipulation method and bulk assay...." to "single cell manipulation using optical tweezers and 96-well format cell adhesion assay...." as recommended. This change provides a more precise description of the methodologies employed.

  1. Discussion (page 13, paragraph 2) - Change "...minimal time required to establish adhesive bonds..." to "...minimal time required to establish adhesion.."

We have adjusted "...minimal time required to establish adhesive bonds..." to "...minimal time required to establish adhesion..." in accordance with the reviewer's suggestion, ensuring terminological accuracy.

  1. Discussion (page 13, paragraph 2) - you write "Moreover, the viability of cells is monitored throughout the process...". Please clarify how you monitor the viability of cells? For example, by monitoring the cell shape?

 We introduced the following sentence in the Discussion section, along with an answer to comment no. 1.

We hope that our responses and revisions address your concerns satisfactorily. Thank you once again for your insightful comments and suggestions, which have helped us improve the quality of our manuscript. We appreciate your time and effort in reviewing our work.

Best regards,

Katarzyna Gdesz-Birula